# Evidences of *Colletotrichum fructicola* Causing Anthracnose on *Passiflora edulis* Sims in China

**DOI:** 10.3390/pathogens11010006

**Published:** 2021-12-22

**Authors:** Wenzhi Li, Fei Ran, Youhua Long, Feixu Mo, Ran Shu, Xianhui Yin

**Affiliations:** Research Center for Engineering Technology of Kiwifruit, Institute of Crop Protection, College of Agriculture, Guizhou University, Guiyang 550025, China; lwz9512@126.com (W.L.); RF18786614232@126.com (F.R.); gs.fxmo19@gzu.edu.cn (F.M.); s17708559157@126.com (R.S.); xhyin@gzu.edu.cn (X.Y.)

**Keywords:** passion fruit, anthracnose, *Colletotrichum fructicola*, first report

## Abstract

Passion fruit (*Passiflora edulis*) is a tropical and subtropical plant that is widely cultivated in China due to its high nutritional value, unique flavor and medicinal properties. In August 2020, typical anthracnose symptoms with light brown and water-soaked lesions on *Passiflora edulis* Sims were observed, which result in severe economic losses. The incidence of this disease was approximately 30%. The pathogens from the infected fruit were isolated and purified by the method of tissue isolation. Morphological observations showed that the colony of isolate BXG-2 was gray to celadon and grew in concentric circles. The orange conidia appeared in the center after 14 days of incubation. The pathogenicity was verified by Koch’s postulates. The internal transcribed spacer (*ITS*), chitin synthase (*CHS-1*), actin (*ACT*), and glyceraldehyde-3-phosphate dehydrogenase (*GAPDH*) were amplified by relevant PCR programs. The multi-gene (*ITS*, *GAPDH*, *ACT*, *CHS-1*) phylogeny analysis confirmed that isolate BXG-2 belongs to *Colletotrichum fructicola*. The inhibitory effect of six synthetic fungicides on the mycelial growth of the pathogen was investigated, among which difenoconazole 10% WG showed the best inhibitory effect against *C. fructicola* with an EC_50_ value of 0.5579 mg·L^−1^. This is the first report of anthracnose on *Passiflora edulis* Sims caused by *Colletotrichum fructicola* in China.

## 1. Introduction

Passion fruit, which belongs to the Passifloraceae family, is a tropical and subtropical plant that contains more than 500 species worldwide [1,2]. Passion fruit are mainly distributed in South America, Central America, Southeast Asia, and China [3,4,5]. The yellow (*Passiflora edulis* f. *flavicarpa*) and purple (*Passiflora edulis* Sims) passion fruit are the main cultivated varieties [6]. Passion fruit is recognized owing to its considerable nutritional value, unique flavor and medicinal properties [7]. The edible parts of the passion fruit are rich in nutrients, such as proteins, sugar, and vitamin C [8]. The pericarp can be used in the production of dietary fiber, pectin, and other bioactive compounds [9]. He et al. found more than 110 phytochemical constituents from different parts of *Passiflora edulis*, among which flavonoids and triterpenoids held the highest levels [10]. Moreover, the seeds are rich in lipids, proteins, minerals, fiber, and bioactive compounds, such as sterols, tocols, carotenoids, and phenolic compounds [11,12]. Therefore, passion fruit is widely used in the pharmaceutical, food, and cosmetic industry.

The production of crops and fruit has been greatly affected by various diseases with the rapid expansion of cultivation areas. Among them, anthracnose is one of the main pathogenic species causing yield loss and the decline of economic values during production, long-distance transport, and cold storage [13]. Studies have shown that the genus *Colletotrichum* is primarily described as causing anthracnose diseases of leaves, stems, flowers, and fruit, as well as crown and stem rots and seedling blight [14]. Based on scientific and economic importance, the genus *Colletotrichum* has been considered to be one of the eight most important plant pathogenic fungi in the world [15]. To date, anthracnose has been reported on many plants. Sun et al. reported the leaf anthracnose caused by *C. spaethianum* on *Hosta ventricosa* [16]. The mango fruit anthracnose caused by *C.*
*asianum* was observed in Indonesia [17]. Uysal et al. reported the causal agent of anthracnose as *C. karstii* on avocado fruits in Turkey [18]. Moreover, postharvest anthracnose caused by *C. fragariae* is one of the most severe diseases of the cultivated strawberry [19].

In August 2020, typical anthracnose symptoms with light brown and water-soaked lesions on passion fruit (*Passiflora edulis* Sims) were observed in an orchard in Zhenfeng, Qianxinan, Guizhou province, China. The disease incidence was approximately 30%, which resulted in severe economic loss and fruit quality decline. Based on this, this study aims to identify the pathogen causing anthracnose by morphology and DNA sequencing. The sequences of the internal transcribed spacer (*ITS*), chitin synthase gene (*CHS-1*), actin gene (*ACT*), and glyceraldehyde-3-phosphate dehydrogenase gene (*GAPDH*) were used for the multi-locus phylogenetic analysis. Additionally, six synthetic fungicides were selected to detect the indoor toxicity against anthracnose caused by *Colletotrichum fructicola*.

## 2. Results

### 2.1. Field Symptoms

The typical anthracnose symptoms in the field were first observed on the surface of passion fruit in August 2020, while there were no symptoms observed on the leaves or stems. At the early stage of the pathogen infection, small water-soaked spots appeared on fruit surface, and then gradually expanded to irregular light yellow lesions (Figure 1). Subsequently, the color changed to dark brown. As a result, the infected passion fruit fell before ripening, causing serious economic losses to local farmers.

### 2.2. Pathogenicity Test

Three isolates of BXG-1, BXG-2, and BXG-3 with different morphological characteristics were obtained from infected passion fruits. The pathogenicity assay showed that the anthracnose symptoms of dark brown sunken necrotic spots were observed 4 days after inoculation using isolate BXG-2 (Figure 2A,B), while the BXG-1 and BXG-3 isolates had no pathogenicity. Subsequently, the pathogenic fungus was re-isolated from the passion fruit that had been infected after inoculation and it exhibited similar morphological characteristics to isolate BXG-2 initially isolated from *Passiflora edulis* Sims fruit. Koch’s postulates confirmed that *Colletotrichum fructicola* was the causal organism of anthracnose on *Passiflora edulis* Sims.

### 2.3. Morphological Identification

The initial colonies of the isolate BXG-2 cultured on PDA medium were white, whereas the old colonies were gray to celadon after 7 days of incubation. The hyphae grew vigorously with neat edges, and grew in concentric circles (Figure 3A,B). After incubation, at 28 °C for 6 days, the diameter of the colony was about 7.2 cm, and the orange conidia appeared in the center after 14 days of incubation (Figure 3C). The conidia were oval or clavate, slightly concave in the center, colorless, and most of which had round ends and contained two oil droplets with the size of (10.5~18.7) μm × (4.2~8.2) μm (av = 15.82 μm × 6.14 μm, n = 100) (Figure 3E). The hyphae were smooth and transparent, with branches (Figure 3F). These morphological characteristics indicated that the BXG-2 isolate belonged to *Colletotrichum* spp.

### 2.4. Phylogenetic Analyses

To identify the isolate BXG-2, the sequences of the *ITS* (549 bp), *GAPDH* (278 bp), *ACT* (268 bp), and *CHS-1* (308 bp) were amplified by PCR and then submitted to GeneBank with the accession numbers of OK178275 (*ITS*), OK256070 (*GAPDH*), OK256068 (*ACT*), and OK256069 (*CHS-1*), respectively. Sequence analysis against BLAST revealed that the *ITS*, *GAPDH*, *ACT*, and *CHS-1* sequences of isolate BXG-2 were 100% similar to *C. fructicola* (*ITS*—MT611204, MT560588, MT393840; *GAPDH*—MN147874, MK836407, MK344241; *ACT*—MN525804, MF039762, MF039761; and *CHS-1*—MN525858, MN525854, MN525852). Moreover, a polygene phylogenetic tree based on *ITS*, *GAPDH*, *ACT*, and *CHS-1* gene sequences showed that the isolate BXG-2 clustered with *Colletotrichum fructicola* (ICMP18581) and was differentiated from other *Colletotrichum* species (Figure 4). Combined with the results of morphological and DNA sequencing, the isolate BXG-2 was confirmed to be *Colletotrichum fructicola*.

### 2.5. Fungicides Screening

The toxicities of six synthetic fungicides against *Colletotrichum fructicola* of *Passiflora edulis* are shown in Table 1. Difenoconazole 10% WG exhibited excellent inhibition effects against *C. fructicola* with the EC_50_ value of 0.5579 mg L^−1^, followed by trifloxystrobin·tebuconazole 75% WG with the EC_50_ value of 1.0354 mg L^−1^, while benzaldehyde·pyraclostrobin 17% SC and fluxapyroxad·pyraclostrobin 42.4% SC showed poor inhibitory effects against the pathogen. These results suggested that difenoconazole 10% WG and trifloxystrobin·tebuconazole 75% WG were effective synthetic fungicides for controlling anthracnose in *Passiflora edulis* caused by *C. fructicola* (Figure 5).

## 3. Discussion

Passion fruit (*Passiflora* ssp.), native to Brazil, is widely cultivated for its high nutritional and medicinal values [20]. The genus *Colletotrichum* has destructive pathogens causing anthracnose diseases of plants. In this study, three isolates (BXG-1, BXG-2, and BXG-3) were obtained from infected passion fruits. Among them, the isolate BXG-2 showed pathogenicity, while the isolates BXG-1 and BXG-3 did not. Given that the morphology of BXG-1 and BXG-3 were different from that of BXG-2, we think that these two non-pathogenetic fungi can be endophytic fungi, which can parasitize in plants without causing any negative symptoms [21]. The conidia of pathogenic isolate BXG-2 were oval or clavate and averaged 15.82 μm × 6.14 μm, which was closely similar to that of *C. fructicola*, reported by Yang et al. for *Pouteria campechiana* [22]. Further multi-locus phylogenetic analyses showed that BXG-2 clustered with *Colletotrichum fructicola*. These results confirmed that the pathogen on the passion fruit was *C**. fructicola*.

Previous studies have demonstrated *C. fructicola* as the causal organism causing anthracnose of fruit, vegetables, and economic crops, including chili [23], pear [24], apple [25], strawberry [26], dragon fruit [27], and tea [28]. Numerous studies have pointed out various ways to control anthracnose caused by *Colletotrichum* spp., including, but not limited to, the application of biorational pesticides, such as essential oil [29], biocontrol strategies [30], synthetic fungicides, and heat treatments [31]. Nowadays, chemical treatments are still the key strategies for controlling plant diseases. To provide effective strategies for controlling anthracnose and reduce the risk of fungicide resistance in *C. fructicola*, we mainly chosen different mixed synthetic fungicide products. Mixed fungicides application is one of the key principles to reduce resistance [32]. Compared with single-site fungicides, the mixed fungicides are beneficial to reduce the risk of pathogen resistance owing to the multiple sites of action [33].

The result we obtained showed that difenoconazole 10% WG exhibited the strongest indoor toxicity against *C. fructicola* with an EC_50_ of 0.5579 mg·L^−1^. Additionally, trifloxystrobin·tebuconazole 75% WG also showed a good inhibition effect with an EC_50_ of 1.0354 mg·L^−1^. Both of them have the potential to prevent the infection of *C. fructicola.* Difenoconazole was considered to be an effective fungicide to control anthracnose development [34]. Shi et al. confirmed the good preventive activity of difenoconazole against *C. fructicola* [35]. In conclusion, this is the first report of *C. fructicola* causing anthracnose on *Passiflora edulis* Sims in China, and the results of this study provide a basis for the field application of benzofenazole and trifloxystrobin·tebuconazole to control anthracnose on passion fruit.

## 4. Materials and Methods

### 4.1. Pathogen Isolation and Purification

The *Passiflora edulis* Sims fruit with typical anthracnose symptoms were sampled from the orchard in Zhenfeng, Qianxinan, Guizhou province, China. The tissues cut from the health and disease junction were firstly cleaned up with sterile distilled water. After air-drying, the tissues were sterilized using 75% (*v*:*v*) ethanol for 20–30 s. Subsequently, the tissues were washed three times with sterile distilled water and placed in the center of potato dextrose agar (PDA) plates. After incubation at 28 °C for 5 d in the dark, the hyphal tips from the mycelia were transferred to new PDA plates and incubated at 28 °C for 7 d. The morphologies of the pure colonies were observed under an optical microscope and then maintained in 30% glycerol and stored at −80 °C until use.

### 4.2. Pathogenicity Assay

Koch’s postulates were used to verify the pathogenicity. In brief, 10 uniform holes (2 mm deep × 1 mm diameter) were made at the passion fruit equator using a sterilized needle, and the holes formed a circle with a diameter of about 7 mm. Then, a disk (7 mm diameter) of pure isolate was inoculated onto the wound. A sterile PDA medium disk was used as a negative control. Subsequently, the passion fruits were placed in a sterile crisper and kept in an artificial climate cabinet (HWS-436, Ningbo Jiangnan Instrument Factory Ltd., Ningbo, China) with a fixed 12 h light/12 h dark photoperiod at 28 °C and 80% relative humidity. The experiment was performed in triplicate. Pathogen isolation was performed again after the appearance of typical symptoms. The re-isolated pathogen was compared with those used for the pathogenicity assay.

### 4.3. Pathogen Identification and Phylogenetic Analyses

The genomic DNA of the pathogen was extracted using the BW-GD2416 Fungal gDNA Isolation Kit (Hangzhou Beiwo Meditech Co., Ltd., Hangzhou, China) according to the manufacturer’s instructions. Primer sequences (listed in Table 2), internal transcribed spacers (*ITS*), the chitin synthase gene (*CHS-1*), actin gene (*ACT*), and glyceraldehyde-3-phosphate dehydrogenase gene (*GAPDH*) were selected with the relevant polymerase chain reaction (PCR) program to amplify the isolate as per Damm et al. [36]. Amplification reactions with the primer pair for each gene were performed in a thermal cycler (T100TM Thermal Cycler, Bio-Rad, Hercules, CA, USA) in a total volume of 20 μL. The *ITS*, *GAPDH*, *ACT*, *CHS-1*, and PCR mixture contained 10 μL of 2 × Taq PCR Master Mix (Shanghai Sangon Biotech Co., Ltd., Shanghai, China), 1 μL of genomic DNA, 1 μL of each primer (10 Μm), and 7 μL of ddH_2_O. The conditions for PCR were as follows: initial denaturation step at 94 °C for 5 min, followed by 34 cycles of denaturation at 94 °C for 30 s, annealing for 30 s with the optimal temperature for each gene (52 °C for *ITS*; 60 °C for *GAPDH*; 58 °C for *ACT*; and 58 °C for *CHS-1*) as per Weir et al. [37], extension at 72 for 45 s, then a final extension for 5 min at 72 °C. Subsequently, the amplified PCR products were sequenced by Sangon Biotech Co., Ltd. (Shanghai, China). The similarity of the resulting DNA sequences was analyzed with the NCBI BLAST tool (available online at www.ncbi.nlm.nih.gov/blast (accessed on 16 March 2021)).Firstly, the highly similar orthologs sequences were downloaded. Subsequently, the multiple alignments of the sequences were performed at https://mafft.cbrc.jp/alignment/server/index.html (accessed on 16 March 2021). The polygene phylogenetic tree was constructed using maximum parsimony analysis in PAUP software based on *ITS*, *GAPDH*, *ACT*, and *CHS-1* gene sequences. Sequences used for polygene phylogenetic tree were listed in Table 3. *Monilochaetes infuscans* CBS869.96 was used as an outgroup. 

### 4.4. Synthetic Fungicides Screening

The mycelial growth rate method was used to determine the indoor toxicity of six synthetic fungicides against *Colletotrichum fructicola*. In brief, the synthetic fungicides were mixed with the PDA medium at a ratio of 1:9 (*v*:*v*), respectively. Subsequently, the PDA medium was emptied into the glass Petri dishes. After the PDA coagulation, a 5 mm diameter mycelial disk divided from a 7 day old colony was respectively placed at the center of the medium with various final concentrations of difenoconazole 10% WG (0.12, 0.37, 1.11, 3.33, and 10.00 mg L^−1^); trifloxystrobin·tebuconazole 75% WG (0.46, 1.39, 4.17, 12.50, and 37.50 mg L^−1^); procyclidine·azoxystrobin 18.7% SE (0.23, 0.69, 2.08, 6.23, and 18.70 mg L^−1^); fluopyram·tebuconazole 35% SC (0.43, 1.30, 3.89, 11.67, and 35.00 mg L^−1^); benzaldehyde·pyraclostrobin 17% SC (0.70, 2.10, 6.30, 18.89, and 56.67 mg L^−1^); and fluxapyroxad·pyraclostrobin 42.4% SC (0.52, 1.57, 4.71, 14.13, and 42.40 mg L^−1^). After incubation for 7 d at 28 °C, the colony diameters were measured in 2 perpendicular directions. Equal amounts of sterile water were used as a negative control. All experiments were performed in triplicate. The inhibitory rate was calculated using the formula: inhibitory rate (%) = [(D_control_ − D_treatment_)/(D_control_ − 5)] × 100, where D represents the diameter of the *C. fructicola* colony.

### 4.5. Statistical Analyses

The data were statistically analyzed using Excel 2010 and DPS 7.05 software. The virulence regression equation (y = ax + b) was obtained by taking the logarithm value of the fungicide concentration (mg L^−1^) as the horizontal coordinate and the biostatistical probability value of the inhibition rate as the vertical coordinate.

## Figures and Tables

**Figure 1 pathogens-11-00006-f001:**
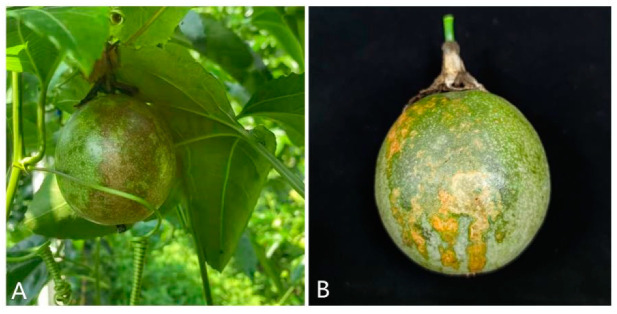
Natural anthracnose symptoms on *Passiflora edulis* caused by *Colletotrichum fructicola.* (**A**) symptoms of early infection and (**B**) symptoms of late infection.

**Figure 2 pathogens-11-00006-f002:**
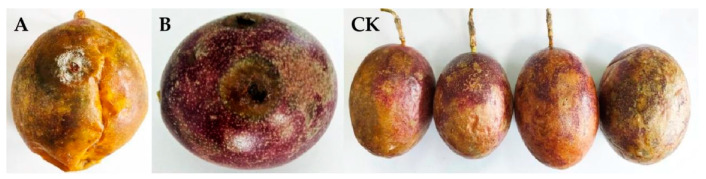
Pathogenicity test results of isolate BXG-2. (**A**,**B**) Puncture inoculation after 4 days and (**CK**) control group after 4 days of inoculation.

**Figure 3 pathogens-11-00006-f003:**
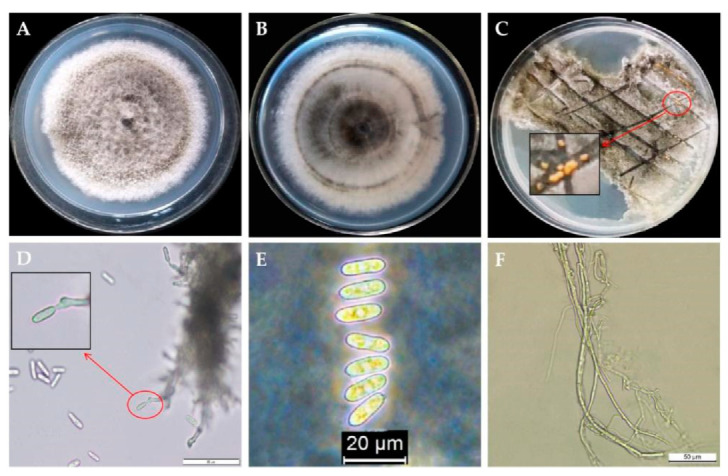
Morphological characteristics of isolate BXG-2. (**A**) Front of BXG-2 colony; (**B**) back of BXG-2 colony; (**C**) orange spore pile of BXG-2; (**D**) sporulation structure of BXG-2, bar = 50 μm; (**E**) spore morphology of BXG-2, bar = 20 μm; and (**F**) mycelial morphology of BXG-2, bar = 50 μm.

**Figure 4 pathogens-11-00006-f004:**
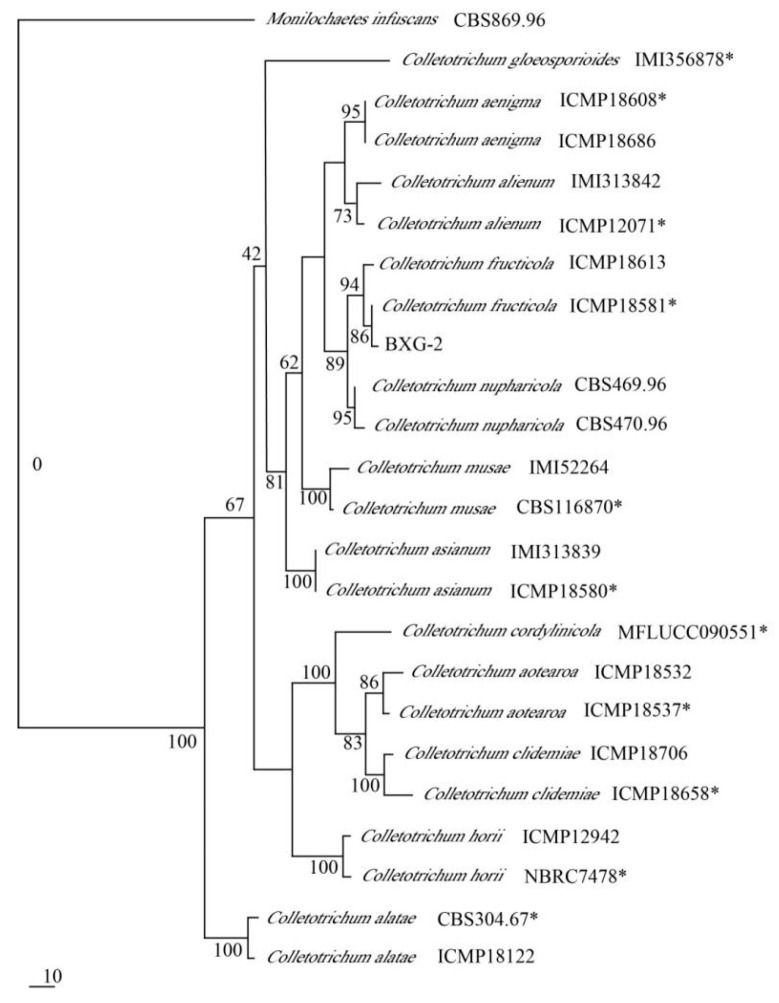
Polygene phylogenetic tree from maximum parsimony analysis based on *ITS*, *GAPDH*, *ACT*, and *CHS-1* gene sequences. ( * = ex-type strains).

**Figure 5 pathogens-11-00006-f005:**
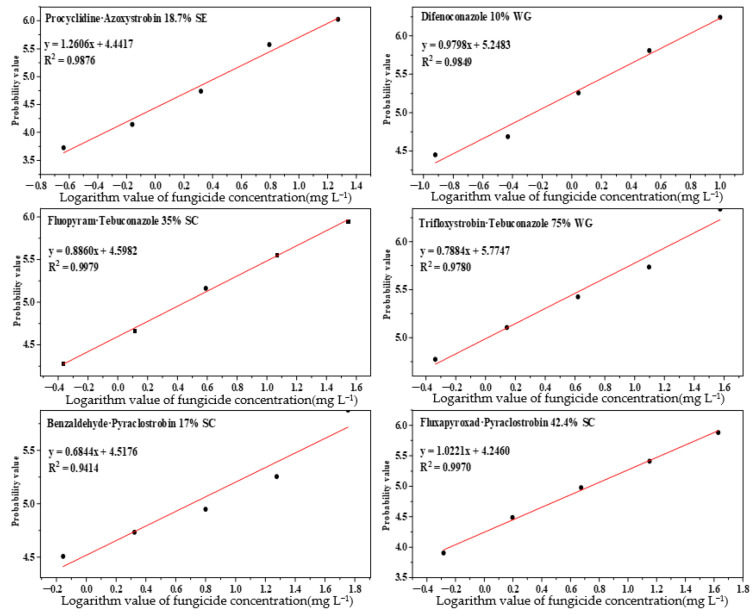
Regression plots of six synthetic fungicides.

**Table 1 pathogens-11-00006-t001:** Indoor toxicity test of six fungicides against *C. fructicola* of *Passiflora edulis.*

Fungicides	Regression Equation of Toxicity	EC_50_ (mg·L^−1^)	Correlation Coefficient (r)
Difenoconazole 10% WG	y = 0.9798x + 5.2483	0.5579	0.9924
Trifloxystrobin·Tebuconazole 75% WG	y = 0.7884x + 5.7747	1.0354	0.9890
Procyclidine·Azoxystrobin 18.7% SE	y = 1.2606x + 4.4417	2.7726	0.9938
Fluopyram·Tebuconazole 35% SC	y = 0.8860x + 4.5982	2.8415	0.9994
Benzaldehyde·Pyraclostrobin 17% SC	y = 0.6844x + 4.5176	5.0687	0.9703
Fluxapyroxad·Pyraclostrobin 42.4% SC	y = 1.0221x + 4.2460	5.4645	0.9985

**Table 2 pathogens-11-00006-t002:** Primers used in this study.

Gene	Primer	Primer Sequences (5′–3′)
*ITS*	ITS1 ITS4	TCCGTAGGTGAACCTGCGG TCCTCCGCTTATTGATATGC
*GAPDH*	GDF GDR	GCCGTCAACGACCCCTTCATTGA GGGTGGAGTCGTACTTGAGCATGT
*ACT*	ACT-512F ACT-783R	ATGTGCAAGGCCGGTTTCGC TACGAGTCCTTCTGGCCCAT
*CHS-1*	CHS-79F CHS-354R	TGGGGCAAGGATGCTTGGAAGAAG TGGAAGAACCATCTGTGAGAGTTG

**Table 3 pathogens-11-00006-t003:** Sequences used for the polygene phylogenetic tree in this study.

Species	Strain No.	GeneBank Accession Number			
		ITS	GAPDH	ACT	CHS-1
*C. aenigma*	ICMP18608 *	JX010244	JX010044	JX009443	JX009774
	ICMP18686	JX010243	JX009913	JX009519	JX009789
*C. alatae*	CBS304.67 *	JX010190	JX009990	JX009471	JX009837
	ICMP18122	JX010191	JX010011	JX009470	JX009846
*C. alienum*	IMI313842	JX010217	JX010018	JX009580	JX009754
	ICMP12071 *	JX010251	JX010028	JX009572	JX009882
*C. aotearoa*	ICMP18532	JX010220	JX009906	JX009544	JX009764
	ICMP18537 *	JX010205	JX010005	JX009564	JX009853
*C. asianum*	IMI313839	JX010192	JX009015	JX009576	JX009753
	ICMP18580 *	FJ972612	JX010053	FJ917506	JX009584
*C. clidemiae*	ICMP18706	JX010274	JX009909	JX009476	JX009777
	ICMP18658 *	JX010265	JX009989	JX009537	JX009877
*C. cordylinicola*	MFLUCC090551 *	JX010226	JX009975	HM470235	JX009864
*C. fructicola*	ICMP18613	JX010167	JX009998	JX009491	JX009772
	ICMP18581 *	JX010165	JX010033	FJ907426	JX009866
*C. gloeosporioides*	IMI356878 *	JX010152	JX010056	JX009531	JX009818
	BXG-2 ^★^	-	-	-	-
*C. horii*	ICMP12942	GQ329687	GQ329685	JX009533	JX009748
	NBRC7478 *	GQ329690	GQ329681	JX009438	JX009752
*C. musae*	IMI52264	JX010142	JX010015	JX009432	JX009815
	CBS116870 *	JX010146	JX010050	JX009433	JX009896
*C. nupharicola*	CBS469.96	JX010189	JX009936	JX009486	JX009834
	CBS470.96 *	JX010187	JX009972	JX009437	JX009835
*M. infuscans*	CBS869.96	JQ005780	JX546612	JQ005843	JQ005801

Note: * = ex-type strains; ^★^ = test strains; and the rest were reference strains.

## Data Availability

The datasets generated and/or analyzed during the study are available from the corresponding author upon reasonable request.

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
