# Peer review of "Evidences of Colletotrichum fructicola Causing Anthracnose on Passiflora edulis Sims in China"

_pathogens, 2021, doi:10.3390/pathogens11010006_

Round 1

Reviewer 1 Report

The manuscript entitled "First report of Colletotrichum fructicola causing anthracnose on Passiflora edulis Sims in China" describes the observation of symptoms on passion fruit consistent with anthracnose, leading to the isolation of three strains of a fungus. One of these three strains was able to produce anthracnose symptoms when detached passion fruit was inoculated. However, the other two strains appeared to be non-pathogenic. More discussion of any potential significance of these two non-pathogen strains is needed in the discussion. However widely applicable are the results likely to be given they are based on a single strain?

The discussion requires improvement. Currently, it repeats some of the content appropriately provided in the introduction, such as the importance of passion fruit and potential impacts of anthracnose. More emphasis should be placed on discussing the results, especially the two non-pathogenic strains. 

How wide spread and/or severe were the symptoms in the orchard from which the pathogen was first isolated? Were plant parts other than fruit impacted? Why were those three strains chosen? Could others have been tested? Was the morphology and/or phylogenetics of the two non-pathogen strains (BXG-1 and BXG-3) examined? 

Why were the particular fungicides chosen? 

In the results, what do "neat", "well developed" and "cotton wool shape" mean when applied to hyphae?

The quality of the photographs is high. Great to see multiple genes sequenced for phylogeny 

Fig 1A,1B (line 72) should be Fig2A, 2B. 

What is strain J-1 (line 97)?

English language and copy editing is required.

Be sure to explain abbreviations before first use e.g. PDA.

Author Response

Comment 1: The manuscript entitled "First report of Colletotrichum fructicola causing anthracnose on Passiflora edulis Sims in China" describes the observation of symptoms on passion fruit consistent with anthracnose, leading to the isolation of three strains of a fungus. One of these three strains was able to produce anthracnose symptoms when detached passion fruit was inoculated. However, the other two strains appeared to be non-pathogenic. More discussion of any potential significance of these two non-pathogen strains is needed in the discussion. However widely applicable are the results likely to be given they are based on a single strain?

Response: We sincerely thank the reviewer for the positive comments and careful reviews!

Reviewer’s comments are extremely constructive and valuable, and very helpful for revising and improving our manuscript. We have studied carefully the comments and have substantial revisions which we sincerely hope meet with approval. In our previous work, three strains (BXG-1, BXG-2 and BXG-3) were obtained from infected passion fruit. Except for BXG-2, neither BXG-1 nor BXG-2 showed pathogenicity. Therefore, we did not do further research on the two non-pathogenic strains. We speculate that they may be endophytic fungi. The importance of non-pathogenic isolates has been highlighted in the discussion section. From the current research results, all the results we got are based on the pathogenic strain BXG-2. More details have been added in Discussion section which marked in blue in the revised manuscript. Thank you most sincerely! (see lines 134-139)

Comment 2: The discussion requires improvement. Currently, it repeats some of the content appropriately provided in the introduction, such as the importance of passion fruit and potential impacts of anthracnose. More emphasis should be placed on discussing the results, especially the two non-pathogenic strains.

 Response: Thank you very much for your good comments and careful reviews! We have removed the duplicate content and disscussed the two non-pathogenic strains. The modified part has been marked in blue font. Thank you most sincerely! (see lines 132-143)

Comment 3: How wide spread and/or severe were the symptoms in the orchard from which the pathogen was first isolated?

Response: Special thanks to you for your good comments and careful reviews! We have mentioned the severe of anthracnose in the introduction section ‘The disease incidence was approximately 30%.’ Thank you most sincerely! (See line 57)

Comment 4:Were plant parts other than fruit impacted?

Response: Special thanks to you for your careful reviews. To date, We didn’t observe any anthracnose symptoms on other plant parts. And we have added a corresponding description which marked in blue in the revised manuscript. Thank you most sincerely! (See line 68)

Comment 5: Why were those three strains chosen? Could others have been tested? Was the morphology and/or phylogenetics of the two non-pathogen strains (BXG-1 and BXG-3) examined?

Response: Thank you very much for your good comments and careful reviews! In our previous study, we isolated 3 kinds of strains with different morphological characteristics which labeled BXG-1, BXG-2 and BXG-3. Therefore, we selected these three types of strains for pathogenicity testing. Given that the morphology of other strains were respectively consistent with the morphology of these three types of strains (BXG-1, BXG-2 and BXG-3), we did not carry out additional pathogenicity tests. From the perspective of pathogenicity, we think that the non-pathogenic strains BXG-1 and BXG-3 are not closely related to this study. They may be endophytic fungi, so we did not examine their morphology and/or phylogenetics. Thank you most sincerely!

Comment 6: Why were the particular fungicides chosen?

Response: Thanks very much for the reviewer's careful reviews on our manuscript! Compared with a single-site fungicide, the mixed fungicides are beneficial to reduce the risk of pathogen resistance owing to the multiple sites of action. Therefore, we we mainly evaluated the indoor toxicity of mixed fungicide products. Thank you most sincerely! (See lines 151-155)

Comment 7: In the results, what do "neat", "well developed" and "cotton wool shape" mean when applied to hyphae?

Response: Special thanks to you for your good comments! We have changed the description of the morphological characteristics of hyphae. Thank you most sincerely! (see lines 91-92)

Comment 8: The quality of the photographs is high. Great to see multiple genes sequenced for phylogeny. Fig 1A,1B (line 72) should be Fig2A, 2B.

Response: Thanks very much for the reviewer's positive comments and careful reviews. ‘Fig 1A, 1B’ has been changed to ‘Fig2A, 2B’. Thank you most sincerely!

Comment 9: What is strain J-1 (line 97)?

Response: Thanks very much for the reviewer's careful reviews on our manuscript! I'm really sorry for the trouble brought to you. I made writing mistakes due to my personal negligence. “J-1” is written incorrectly, it is actually “BXG-2”. We have made corrections. Thank you most sincerely! (see line 104)

Comment 10: English language and copy editing is required.

Response: Thank you very much for your good advice! We have carefully checked the details in the article and made reasonable changes. Thank you most sincerely!

Comment 11: Be sure to explain abbreviations before first use e.g. PDA.

Response: Thanks very much for the reviewer's careful comment to our manuscript! We have carefully checked the abbreviations in the article and explained them accordingly. Thank you most sincerely!

Reviewer 2 Report

Review “First report of Colletotrichum fructicola causing anthracnose on Passiflora edulis Sims in China”

I enjoyed reading this manuscript. I think it presents the authors’ findings in a clear and interesting way. However, there are important sections that should be addressed. Based on the following comments, my recommendation is major revisions: 

1.- I think the title should be changed. This work is important, but indeed it is not the first time that the genus Colletotrichum is found on Passiflora edulis. Actually, the authors indicated that the genus Colletotrichum has been reported in Passiflora edulis since 2003. One suggestion is to change the title to: “Evidences of Colletotrichum fructicola causing anthracnose on Passiflora edulis Sims in China” and to focus the discussion on the severity of the symptoms, the identification of the pathogen causing them (discarding BXG-1 and BXG-3 and why) and the feasibility of the use of the fungicide proposed.

2.- Please change “strains” to “isolates” as these fungi were isolated from the infected plant tissues.

3.- The authors mentioned the “strains” (please change to “isolates”) BXG-1 and BXG-3 were not pathogenic. Is it possible that they lost their pathogenicity after being cultured on PDA? Please, clarify this on the Discussion section. The Discussion is short, please add more information on Colletotrichum fructicola and how it is managed in other crops.

4.- What is strain J-1? I could not find a description of it or how it was obtained or why its sequences were used.

Author Response

Comment 1: I enjoyed reading this manuscript. I think it presents the authors’ findings in a clear and interesting way.

Response: Thanks very much for the reviewer's positive reviews! Thank you most sincerely!

Comment 2: I think the title should be changed. This work is important, but indeed it is not the first time that the genus Colletotrichum is found on Passiflora edulis. Actually, the authors indicated that the genus Colletotrichum has been reported in Passiflora edulis since 2003. One suggestion is to change the title to: “Evidences of Colletotrichum fructicola causing anthracnose on Passiflora edulis Sims in China” and to focus the discussion on the severity of the symptoms, the identification of the pathogen causing them (discarding BXG-1 and BXG-3 and why) and the feasibility of the use of the fungicide proposed.

Response: We sincerely thank for the reviewer's careful and constructive comment to our manuscript! According to your suggestion, we have changed the title to: “Evidences of Colletotrichum fructicola causing anthracnose on Passiflora edulis Sims in China”. We have also added discriptions on the severity of the symptoms and the feasibility of the use of the fungicides in discussion section. In our previous work, three isolates BXG-1, BXG-2 and BXG-3 were obtained from infected passion fruit, neither BXG-1 nor BXG-2 showed pathogenicity. We speculated that isolates BXG-1 and BXG-2 may be endophytic fungi. Thus, we did not do further research on the two non-pathogenic strains. Thank you most sincerely! (see lines 132-155)

Comment 3: Please change “strains” to “isolates” as these fungi were isolated from the infected plant tissues.

Response: Thanks very much for the reviewer's constructive reviews on our manuscript! We have changed “strains” to “isolates” in the revised manuscript. Thank you most sincerely!

Comment 4: The authors mentioned the “strains” (please change to “isolates”) BXG-1 and BXG-3 were not pathogenic. Is it possible that they lost their pathogenicity after being cultured on PDA? Please, clarify this on the Discussion section. The Discussion is short, please add more information on Colletotrichum fructicola and how it is managed in other crops.

Response: Thanks very much for the reviewer's careful reviews on our manuscript! Actually, the morphological characteristics of isolates BXG-1 and BXG-3 were obviously different from that of BXG-2. We think that BXG-1 and BXG-3 may be endophytic fungi. Further disscussion has been added and marked in blue in the revised manuscript. Thank you most sincerely! (see lines 132-155)

Comment 5: What is strain J-1? I could not find a description of it or how it was obtained or why its sequences were used.

Response: Thanks very much for the reviewer's careful reviews on our manuscript! I'm really sorry for the trouble brought to you. I made writing mistakes due to my personal negligence. “J-1” is written incorrectly, it is actually “BXG-2”. We have made corrections. Thank you most sincerely! (see line 104)

Reviewer 3 Report

In this study, Li and co-workers present evidence that the fungus Colletotrichum fructicola is the cause of anthracnose disease in passion fruit of Chinese orchards. They do so by examining field symptoms and performing pathogenicity tests (Koch's postulates). They also provide sequence data, extracted from the fungus isolated from symptomatic fruit and evaluate the effects of several fungicides against the identified pathogen in vitro.

In general, it is a well-written study, compact, and methodologically justified. It should also be of significant interest, as it involves an edible product of economic importance. My main remarks concern the presentation of the results:

  1. It is not clear whether Fig.1 depicts the "naturally" infected fruit or the inoculated ones (see Line 65 vs L. 72).
  2. It seems that EC50 estimates were deduced graphically. Thus, it would be appropriate to present regression plots, at least as supplementary figures.
  3. "Biostatistical probability value of inhibition rate" was the proxy for fungicide evaluations. However, no definition is provided nor information on potential background effect (controls) and how this was treated is given.
  4. In Line 106, the claim about "similarity of 86 %" must be erroneous, as values on the nodes of phylogenetic trees should represent either bootstrap or jackknife support scores and not sequence similarity. Please, clarify this and give more details on the phylogenetic tree reconstruction in Methods.

Some minor comments:

Line 12: Pls remove [space] before comma.

L. 38-40. This paragraph should be about anthracnose. These first three lines can be misleading that C. fructicola is already proven to cause anthracnose in passion fruit. Thus, they should be placed in the previous paragraph.

L. 43: The phrase "Anthracnose is the main pathogenic species of various crops" should be revised.

L. 76: Passiflora edulis should be in italics.

L. 97: What is the strain J-1? It is never mentioned before or after.

L. 121: Pls, make p capital in passiflora edulis.

L. 205. Pls, replace date by data.

Author Response

Comment 1: In this study, Li and co-workers present evidence that the fungus Colletotrichum fructicola is the cause of anthracnose disease in passion fruit of Chinese orchards. They do so by examining field symptoms and performing pathogenicity tests (Koch's postulates). They also provide sequence data, extracted from the fungus isolated from symptomatic fruit and evaluate the effects of several fungicides against the identified pathogen in vitro.

In general, it is a well-written study, compact, and methodologically justified. It should also be of significant interest, as it involves an edible product of economic importance.

Response: Thanks very much for the reviewer's positive reviews on our manuscript! We also sincerely thank reviewer very much for giving us an opportunity to revise our manuscript! Thank you most sincerely!

Comment 2: It is not clear whether Fig.1 depicts the "naturally" infected fruit or the inoculated ones (see Line 65 vs L. 72).

Response: Thanks very much for the reviewer's careful reviews on our manuscript! We have added more detailed descriptions on the naturally infected fruit. Thank you most sincerely! (see lines 74-75)

Comment 3: It seems that EC50 estimates were deduced graphically. Thus, it would be appropriate to present regression plots, at least as supplementary figures.

Response: We sincerely thank the reviewer for the careful reviews! Reviewer's comments are extremely constructive and valuable, and very helpful for revising and improving our manuscript. We have added the corresponding regression plots (see Figure 4). Thank you most sincerely! (see lines 129-130)

Comment 4: "Biostatistical probability value of inhibition rate" was the proxy for fungicide evaluations. However, no definition is provided nor information on potential background effect (controls) and how this was treated is given.

Response: We sincerely thank the reviewer for the careful reviews to our manuscript! We have added the formula for mycelial growth inhibition rate which marked in blue in the revised manuscript. Thank you most sincerely! (see lines 213-215)

Comment 5: In Line 106, the claim about "similarity of 86 %" must be erroneous, as values on the nodes of phylogenetic trees should represent either bootstrap or jackknife support scores and not sequence similarity. Please, clarify this and give more details on the phylogenetic tree reconstruction in Methods.

Response: We sincerely thank the reviewer for the careful reviews and warm work to our work! The incorrect description "similarity of 86 %" has been removed. We changed the original description to: "the isolate BXG-2 clustered with Colletotrichum fructicola (ICMP18581) and was differentiated from other Colletotrichum species". More details on the phylogenetic tree reconstruction in Methods has been added. Thank you most sincerely! (see line 112-113)

Comment 6:Line 12: Pls remove [space] before comma.

Response: Thanks very much for the reviewer's careful reviews on our manuscript! The [space] before comma has been removed. Thank you most sincerely! (see line 110)

Comment 7: L. 38-40. This paragraph should be about anthracnose. These first three lines can be misleading that C. fructicola is already proven to cause anthracnose in passion fruit. Thus, they should be placed in the previous paragraph.

Response: Thanks very much for the reviewer's careful reviews on our manuscript! These first three lines has been placed in the previous paragraph. In addition, We also added more discriptions about anthracnose. Thank you most sincerely! (see lines 29-30)

Comment 8: L.76: Passiflora edulis should be in italics.

Response: Special thanks to you for your careful comments! All the "Passiflora edulis" in this article have been italicized. Thank you most sincerely!

Comment 9: What is the strain J-1? It is never mentioned before or after.

Response: Thanks very much for the reviewer's careful reviews on our manuscript! I'm really sorry for the trouble brought to you. I made writing mistakes due to my personal negligence. "J-1" is written incorrectly, it is actually "BXG-2". We have made corrections. Thank you most sincerely! (see line 104)

Comment 10: L. 121: Pls, make p capital in passiflora edulis.

Response: Thanks very much for the reviewer's careful reviews on our manuscript! We have made p capital in passiflora edulis. Thank you most sincerely!

Comment 11: Pls, replace date by data.

Response: Special thanks to you for your careful comments! The "date" has been changed to " data". Thank you most sincerely! (see line 233)

Round 2

Reviewer 2 Report

I thank the authors for their responses. This manuscript can be accepted after a final careful check of the cited and listed references. Also, I strongly recommend the authors to check for any typographical error that the text might still have.